# Characteristics of Waste Iron Powder as a Fine Filler in a High-Calcium Fly Ash Geopolymer

**DOI:** 10.3390/ma14102515

**Published:** 2021-05-12

**Authors:** Toon Nongnuang, Peerapong Jitsangiam, Ubolluk Rattanasak, Weerachart Tangchirapat, Teewara Suwan, Suriyah Thongmunee

**Affiliations:** 1Graduate Program in Civil Engineering, Department of Civil Engineering, Faculty of Engineering, Chiang Mai University, Huay Kaew Road, Muang, Chiang Mai 50200, Thailand; toon_n@cmu.ac.th; 2Center of Excellence in Natural Disaster Management, Department of Civil Engineering, Faculty of Engineering, Chiang Mai University, Chiang Mai 50200, Thailand; teewara.s@cmu.ac.th (T.S.); suriyah.t@cmu.ac.th (S.T.); 3Department of Chemistry, Faculty of Science, Burapha University, Chonburi 20131, Thailand; ubolluk@go.buu.ac.th; 4Construction Innovations and Future Infrastructure Research Center (CIFIR), Department of Civil Engineering, Faculty of Engineering, King Mongkut’s University of Technology Thonburi, Bangkok 10140, Thailand; weerachart.tan@kmutt.ac.th

**Keywords:** geopolymer, geopolymer paste, waste iron powder, composite material

## Abstract

Geopolymer (GP) has been applied as an environmentally-friendly construction material in recent years. Many pozzolanic wastes, such as fly ash (FA) and bottom ash, are commonly used as source materials for synthesizing geopolymer. Nonetheless, many non-pozzolanic wastes are often applied in the field of civil engineering, including waste iron powder (WIP). WIPs are massively produced as by-products from iron and steel industries, and the production rate increases every year. As an iron-based material, WIP has properties of heat induction and restoration, which can enhance the heat curing process of GP. Therefore, this study aimed to utilize WIP in high-calcium FA geopolymer to develop a new type of geopolymer and examine its properties compared to the conventional geopolymer. Scanning electron microscopy and X-ray diffraction were performed on the geopolymers. Mechanical properties, including compressive strength and flexural strength, were also determined. In addition, setting time and temperature monitoring during the heat curing process were carried out. The results indicated that the addition of WIP in FA geopolymer decreased the compressive strength, owing to the formation of tetrahydroxoferrate (II) sodium or Na_2_[Fe(OH)_4_]. However, a significant increase in the flexural strength of GP with WIP addition was detected. A flexural strength of 8.5 MPa was achieved by a 28-day sample with 20% of WIP addition, nearly three times higher than that of control.

## 1. Introduction

Due to global warming, conventional concrete in construction results in tons of carbon dioxide (CO_2_) emissions to the atmosphere, leading to the greenhouse effect. For decades, eco-friendly materials have been widely used in civil engineering work worldwide. By-products or waste materials are being investigated as alternatives in civil construction, particularly as replacements for cement [1,2,3], because they can provide properties of raw materials and effectively reduce construction costs [4,5,6]. These attempts aim to achieve three of the UN Sustainable Development Goals (SDGs)—sustainable cities and communities (11), responsible consumption and production (12), and climate action (13). One environmentally-friendly cementitious material is geopolymer, which can partially or fully replace Portland cement, emitting significantly less CO2 [7]. It can utilize any aluminosilicate by-products by incorporating alkaline solutions. Coal fly ash (FA) is a pozzolanic material that is a by-product of a coal-fired power plant and widely used as a source material for geopolymer production [8,9,10,11]. Geopolymerization is a result of an inorganic polycondensation reaction from the aluminosilicate binders with an alkaline solution [12], which requires heat curing from an external energy source to activate the reaction and accelerate the strength development [13]. Typically, an oven is applied at a temperature ranging from 40 to 90°C for 4 to 48 h in the heat curing process [14,15]. Geopolymer could be made as geopolymer paste, mortar, or composites. Several researchers tried using additional materials, such as steel fiber [16], polyvinyl alcohol fiber (PVA) [17], and even natural fiber like hemp [18], to reinforce geopolymer properties.

Researchers have report the use of iron-rich fly ash in geopolymer, which this fly ash was classified by air classifier [19]. Results showed that iron compound in fly ash resulted in the positive effect on geopolymer properties. In addition, the high iron content in the staring material for the geopolymer synthesis can lead to the formation of ferro–silico-aluminate geopolymer in which some Al atom is substitute by Fe atom [20,21]. However, production temperature of this geopolymer type is around 600 °C–800 °C [20]. The behavior of iron (Fe^3+^) during geopolymerization reaction relies on its chemical and mineralogical state in the starting materials [21]. Low-iron ash has been suggested to use in geopolymer since the harmful action of source ferrous (Fe^2+^) compound can block the development of geopolymerization reaction [20]. Therefore, using of iron-rich ash in geopolymer is considered. Since there are many waste materials produced from civil engineering applications, such as waste iron powder (WIP). In Thailand, iron and steel industries produce more than 2M tons of waste iron and steel powders annually [22], and this rate is growing, creating a long-term environmental problem for the future. Some recent works applied the WIP for mixing in concrete [23] or asphalt concrete [24,25]. However, the application of WIP in geopolymer has not been reported.

Based on the outstanding properties of iron powder (i.e., good heat conduction), the WIP utilization in geopolymer can benefit from the mechanical properties and the heat transfer in oven curing. The heat energy penetrates the material’s surface and transfers to the inner core by thermal convection; nevertheless, geopolymerization may take place at a slow rate, owing to the gentle thermal gradients inside the material [26]. WIP has better thermal properties than FA, such as heat conduction and heat retention. Consequently, proper WIP content in geopolymer can possibly contribute to the heat transfer.

This research is novelty which is the unique application of an iron powder as a filler in geopolymer, particularly WIP. Therefore, this study aims to discover the potential of WIP utilization in high-calcium FA geopolymer in comparison to conventional FA geopolymer through the physical and mechanical properties and microstructure. The use of the waste material is always a challenge in the civil engineering work field.

## 2. Materials and Methodology

### 2.1. Materials

#### 2.1.1. Fly Ash

Fly ash (FA) from the Mae Moh lignite-coal–fired power plant in Lampang, Thailand, was used as source material for geopolymer production. Energy-dispersive X-ray fluorescence spectrometer (EDXRF, JSX3400R, JEOL Ltd., Tokyo, Japan) analysis and scanning electron microscopy (SEM, JSM-5910LV, JEOL Ltd., Tokyo, Japan) were performed to investigate the chemical compositions and the microstructure. The results of EDXRF and SEM are shown in Table 1 and Figure 1, respectively. FA comprises spherical particles due to the high combustion temperature of coal, leading to the sintering of ash particles and resulting in the spherical shape [10]. The median diameter (*D_50_*) of particles was 19 µm. According to ASTM C618 [27], this FA was classified as type C FA because of the excessive amount of sulfur trioxide (SO_3_, more than 5%) and more than 20% lime (CaO) and, therefore, could be defined as a high-calcium FA. The CaO content of 31.41% in the FA was due to the low-grade lignite coal. The calcium content of this lignite coal is rather high and increased with the depth of coal mining [28]. Iron oxide (Fe_2_O_3_) was also high, resulting reddish-brown color. The FA in this study had a specific gravity of 2.3.

#### 2.1.2. Waste Iron Powder

The WIP in this study was a waste material from a bearing manufacturer in Thailand. Because of the industrial process, WIP was contaminated with oil, which needed to be removed with detergent and water. The particle size of WIP was analyzed by particle size analyzer showing the median size (*D_50_*) of 22 µm. The chemical compositions by EDXRF analysis and the microstructure observation by SEM are also presented in Table 1 and Figure 1, respectively. The specific gravity of this WIP was 7.2.

#### 2.1.3. Alkaline Activators

The alkaline activators used in this research were sodium hydroxide (NaOH, SH) and sodium silicate (Na_2_SiO_3,_ SS) solutions. A laboratory-grade NaOH with 99% purity was obtained from ACI Labscan Ltd., Thailand. For the preliminary study, using of high NaOH concentration (10 M) with high-calcium fly ash resulted in the fast setting of paste due to the reaction between NaOH and CaO. Therefore, lower NaOH concentration of 8 M was selected to overcome this problem and provide the more workability of paste. Solid NaOH was dissolved in the distilled water to obtain the NaOH concentrations of 4, 6, and 8 M. A commercial-grade Na_2_SiO_3_ solution was purchased from a local distributor from Kusawad Chemical Group Ltd., Chiang Mai, Thailand with concentrations of 27 wt % SiO_2_ and 8 wt % Na_2_O. The alkaline activators were the blended solutions of Na_2_SiO_3_and NaOH, yielding weight ratios (SS/SH) of 0.67 and 1.00.

### 2.2. Experimental Procedures

#### 2.2.1. Manufacturing Process

First, FA and WIP were dried in the oven for 24 h before mixing. For the alkaline activator, Na_2_SiO_3_ and the desired concentration of NaOH were blended to obtain SS/SH values of 0.67 and 1.00. In the mixing process, FA was combined with the blended alkaline solution using a blender. The mass ratio of liquid and binder (L/B) used in the test was 0.45. Next, the fresh geopolymer paste was poured in 4 × 4 × 16 cm^3^ prisms and was left at room temperature for 1 h. All samples were then placed in the oven for heat curing at the desired temperature between 30 °C–60 °C for 24 h as shown in Table 2. The specimens were cooled down to room temperature and wrapped with a cling film after de-molding. All specimens were then cured at ambient temperature (25 °C) until they reached their testing age. The details of mixing proportion are summarized as follows.

#### 2.2.2. Mixing Proportion

Controlled geopolymer paste (CGP).

The controlled geopolymer consisted of FA, NaOH, and Na_2_SiO_3_ solution. The oven-dried FA was mixed with the alkaline activators, which are the blended solution of 4, 6, and 8 M NaOH and Na_2_SiO_3_ solution with the mass SS/SH values of 0.67 and 1.00 in order to produce the high strength geopolymer [14].

WIP mixed geopolymer mortar (WGm).

Amounts of 5%, 10%, 15%, and 20% of WIP were added to the geopolymer paste as a fine filler by %weight of binder (FA). The details of the mixing process are given in Table 2. ‘WxGP’ indicates WIP mixed geopolymer paste, where x is the percent mass of WIP.

#### 2.2.3. Analytical Methods

Physical properties.

The setting times were tested using Standard Vicat apparatus (L0028/A, Controls Group, Milan, Italy) in accordance with ASTM C191-19 [29]. The moment the raw material made contact with the alkaline activators was defined as the start of the setting time test. The bulk density of hardened geopolymers was measured at 28 days prior to the mechanical properties test by measuring a prismatic sample’s dimensions and weight.

Mechanical properties.

The mechanical properties of hardened CGPs and WGPs, including compressive strength and flexural strength, were carried out from the three and 28 day–aged samples using the CST (Compression machine, CST Instruments (Thailand) Part. Ltd., Thailand) compression machine in accordance with the BS EN 196-1 [30]. The flexural strength test was conducted on 4 × 4 × 16 cm^3^ prismatic samples using a three-point loading method in the 100-mm span length of two supporting points. The parts obtained from the flexural strength test were then tested for compressive strength at a loading rate of 1.0 mm/min and 4 × 4 cm^2^ loading area.

#### 2.2.4. Microstructure Analysis

The X-ray Diffraction (XRD, Rigaku SmartLab, Rigaku Corporation., Tokyo, Japan) and SEM techniques (JSM-5910LV, JEOL Ltd., Tokyo, Japan) were performed to observe the crystalline phases and microstructure. The energy-dispersive X-ray spectroscopy (EDX, JSM-5910LV, JEOL Ltd., Tokyo, Japan) technique was used to identify the chemical compositions of products in the spectrum.

#### 2.2.5. Temperature Monitoring

To consider the thermal properties of W10GP sample, the high-calcium FA geopolymer with 10% WIP sample was set up for temperature monitoring using type K thermocouples during the 24-h heat curing process (60 °C in the oven) and for three hours after the oven curing. The thermocouples were placed to measure temperature at three locations: surface, core, and oven (ambient).

## 3. Results and Discussion

### 3.1. Physical Properties

#### 3.1.1. Bulk Density of Hardened Geopolymer Paste

Figure 2 shows the bulk densities of the hardened geopolymers pastes with varying WIP contents. The WIP content clearly affected the density of geopolymer, owing to WIP’s significantly greater specific gravity (7.2) compared to FA (2.3). The bulk density increased with WIP content.

#### 3.1.2. Setting Time Test

Setting time is a defining characteristic of geopolymer. The hardening process is related to the formation of calcium aluminosilicate hydrate gel (C–A–S–H) and Ca(OH)_2_ depending on the raw materials and alkaline activators [28,31]. The high-calcium FA geopolymer had a short setting time because of the high CaO content in FA. In accordance with Figure 3, setting time tests were performed using Vicat apparatus. The results showed that WGP samples had both lower initial and final setting times than the CGP samples, in which hematite (iron(III) oxide, Fe_2_O_3_) in WIP does not contribute to the geopolymer’s reaction or hydration. The initial and final setting times for CGP were 576 and 1020 s, respectively; meanwhile, the initial setting time of the WGP samples was 338 to 368 s, and the final setting time was 604 to 746 s. The addition of WIP reduced the setting time of the WGP samples, owing to the reaction between WIP and alkali solution, and increase in solid content with same alkali volume. The new product was formed in the system and accelerated the hardening of geopolymer. The reaction between WIP and alkali solutions could form the iron(II) oxide or ferrous oxide (Fe(OH)_2_), as shown in Equation (1).
Fe(s) + 2NaOH(aq) → Fe(OH)_2_(s) + 2Na^+^(aq)(1)

Comparison between samples, the higher WIP contents in the mixtures resulted in the longer initial setting times, which can be due to a hindrance of FA-alkali reaction by WIP particles. In contrast, the final setting time of WGP was reduced by adding more WIP, likely owing to the formation of Fe(OH)_2_. In summary, the addition of WIP promoted a quicker hardening process of geopolymer.

### 3.2. Mechanical Properties

Figure 4a illustrates the compressive strength of samples mixed with 8 M NaOH alkaline activator with SS/SH of 1.00 at a curing temperature of 60 °C. CGP samples yielded a high early strength of 46.6 MPa at three-day age, and the strength developed with time because of the geopolymerization and pozzolanic reactions. The strength reached 66.4 MPa at 28-day curing time. The early strength of some WGP samples resulted in greater compressive strength than that of CGP. For the three-day age, W5GP obtained a compressive strength of 44.6 MPa, while W20GP achieved 59.9 MPa. This difference could be because the reaction between Fe(OH)_2_ and NaOH solutions led to the formation of tetrahydroxoferrate(II) sodium, Na_2_[Fe(OH)_4_], which can occur under heat curing, as shown in Equation (2). In addition, Ca(OH)_2_ can be formed, owing to the reaction between CaO in FA with alkali solution.
Fe(OH)_2_(s) + 2NaOH(aq) → Na_2_[Fe(OH)_4_](s)(2)

However, the compressive strength of the WGP group noticeably decreased in comparison to CGP at 28-day curing time. The compressive strength wildly dropped down to around 48.5 MPa, 76.0% of CGP’s strength. The formation of both Na_2_[Fe(OH)_4_] and Ca(OH)_2_ may have had an adverse effect on the strength of a sample at longer age.

The results of the flexural strength test are displayed in Figure 4b. The flexural strength of CGP was 3.8 and 3.0 MPa at 3- and 28-day curing times, respectively. The development of flexural strength of the WGP group was similar to that of compressive strength. The 28-day samples provided lower flexural strength than the three-day samples. Nevertheless, the flexural strengths of WGP samples were considerably higher than those of the CGP, especially for the 28-day W20GP sample with a flexural strength of 8.5 MPa, almost three times that of the conventional geopolymer.

The addition of WIP as a filler is a new approach to making geopolymer paste. However, an optimum content can effectively contribute to the best mechanical performance for geopolymer. The results of these tests revealed that the WGP had lower compressive strength but significantly higher flexural strength in comparison to the controlled geopolymer paste. The WIP had an adverse effect on the compressive strength since the reaction between micro-WIP particles and alkali solution occurred. WIP was the waste and had an impurity which could affect the geopolymerization reaction as the compressive strength of WIP geopolymer was lower than that of control. However, flexural strength had much improved due to the grain-shape of WIP.

For only WGP samples, higher amounts of WIP mixed in the geopolymer improved both compressive strength and flexural strength. The compressive strength of W5GP was 32.2 MPa and rose to 50.5 MPa in the W20GP (20% content of WIP) sample, 76% of the CGP strength. In a similar way, the flexural strength of W5GP was 2.6 MPa and significantly increased to 5.0, 4.7, and 8.5 MPa in the W10GP, W15GP, and W20GP samples, respectively. Hence, addition of WIP can reinforce the geopolymer matrix by long grain shape and rough surface of WIP, resulting in the higher flexural strength.

### 3.3. NaOH Concentration and SS/SH

The previous research of Rattanasak [32] showed that a higher concentration of NaOH and larger amount of Na_2_SiO_3_ led to higher compressive strength in geopolymer because the high concentration of NaOH could dissolve more Si^3+^ and Al^4+^ ions out of the surface of FA particles to form the geopolymerization gel, connecting each FA particle. Moreover, a large amount of Na_2_SiO_3_solution added more silica to form the gel in geopolymer. To prove this theory, this research also studied the behavior of WGP on the NaOH concentration and SS/SH, and the results are illustrated in Figure 5 and Figure 6. Figure 5 shows that both compressive strength and flexural strength of W20GP increased along with the higher concentration of NaOH. Furthermore, Figure 6 shows the results between two mixtures of geopolymer paste with the different SS/SH values and NaOH concentrations. The higher compressive strength was obtained from the samples with an SS/SH value of 1.00 since Na_2_SiO_3_ provided the dissolved SiO_2_ to the system, resulting in more formations of C–S–H and aluminosilicate compound in geopolymer. However, both SS/SH values of 0.67 and 1.00 yielded the same level of flexural strength. In summary, WIP mixed with FA-based geopolymer paste behaved similarly to a conventional geopolymer.

### 3.4. Heat Curing Temperature

Figure 7 shows the heat curing temperature affected the W20GP samples, varying the temperature from 30 °C–60 °C. It was seen that oven curing resulted in higher strength of geopolymer. This result corresponded to the typical heat curing temperature of FA-based geopolymer [14]. A heat curing temperature range of 40 °C–75 °C could provide the heat for the formation of geopolymerization adequately. However, extremely high heat curing (over 80 °C) could negatively affect the geopolymer properties, owing to a quick loss of moisture content and the addition of micro-cavities in the structure [33]. However, when WIP was applied in geopolymer mixture, an internal temperature of sample could be higher than oven temperature due to the heat conduction of WIP. Therefore, a curing temperature of 60 °C reduced the strength of material.

In summary, no remarkable difference in the appropriate temperature for heat curing between WGP and conventional FA-based geopolymer was found.

### 3.5. Temperature Monitoring

Figure 8a illustrates the heat monitoring result for the 24-h heat curing process. Type K thermocouples were installed to measure the temperature at three locations: sample core, sample surface, and oven. The temperature measurement presented no apparent difference between the core and surface temperatures in both groups of CGP and W10GP; therefore, the result is only displayed in terms of core temperature. According to the results, the temperature increased to 60 °C after four to five hours in the heat curing process; hence, the heat transfer theoretically occurred by thermal convection [34]. Nevertheless, the addition of 10% WIP into the FA-based geopolymer had no impact on temperature during the heat curing process because of the discontinuous iron matrix in geopolymer mass. However, Figure 8b shows the temperature in the three hours after oven curing. The W10GP’s temperature clearly lasted longer than the CGP’s. Therefore, WIP played a significant role in heat retention after the heat curing process and provided the longer heat curing time under this heat holding situation.

### 3.6. Morphological Analysis

The results of XRD patterns of CGP and WGP mixtures are given in Figure 9. At the testing age of 28 days, the major phase of each group was amorphous, indicating a broad hump at the region of 20°–36° 2θ. The phases mainly contained kyanite (Al_2_O_5_Si), tobermorite (Ca_3_HO_9_Si_3_), portlandite (Ca(OH)_2_), and magnetite (Fe_3_O_4_) in all groups. The peaks at 43°–45° 2θ between CGP and WGP, which were of iron (Fe) and nepheline zeolite, were notably different. Sodium iron(III) oxide was also detected in the specimen, owing to the reaction between WIP and alkali solution under heat curing. The addition of WIP in samples mainly increased the peaks of Fe. Particularly, the peak area from 20° to 40° 2θ seemed to reduce, which implied that WIP acted as a filler and no geopolymeric product was found in this system. Origin program was used to calculate the hump area under the peak between 20 and 37° 2θ; the results are shown in Table 3.

SEM and EDX techniques were used to investigate the microstructure of the 28 day-aged WGP mixed with a solution of 8 M NaOH and SS/SH value of 1.00, as illustrated in Figure 10 and Figure 11. Figure 10 displays the appearances of an amorphous structure with some unreacted FA particles surrounded by geopolymeric gel. The residual shells of FA after the reaction are clearly seen in the geopolymer matrix. Large amounts of amorphous network gel product and CSH gel were generated, covering the residual particles to form a continuous gel mass and resulting in a relatively dense microstructure [35,36]. The EDX spectrum revealed high contents of Si, Al, and Ca, which were the products of hydration and geopolymeric reaction. There were small clusters of Fe scattering around the geopolymeric gel. The EDX spectrum also showed the high peak of Fe, which corresponded to the addition of WIP in the geopolymer paste. Moreover, in the higher WIP content samples, many large clusters of Fe were settled.

The SEM results showed a major group of Fe coated with C–A–S–H gel and geopolymeric gel, as illustrated in Figure 11. The flat and twisted iron particles were evident. A gel formation of geopolymerization and hydration connected these uniquely shaped particles together, leading to hermetical iron clusters that had a strong interlocking and bonding between the iron particles. These clusters contributed to the resistance of flexural load in the samples. Therefore, adding more WIP content significantly promoted the flexural strength in the geopolymer.

Nevertheless, the reduction in compressive strength must be explained. The main component of Fe in WIP reacted with the high alkaline solution. The product from this reaction was Na_2_[Fe(OH)_4_] under heat curing. The occurrence of Na_2_[Fe(OH)_4_] increased in the volume of iron compounds, causing stresses in the geopolymer matrix [37,38,39]. Zeolite could form when an SS/SH of 1.0 was used in the mixture [40]. After that, the microcracks occurred in the structure, resulting in the reduction of compressive strength and flexural strength in the passing time. The presence of iron compound in the microstructure. These crystalline expansions weakened the completion of the geopolymer structure, according to XRD analysis. The crystalline phases of kyanite, portlandite, and C–S–H gel of tobermorite, which were the main products of geopolymerization and pozzolanic reaction, were reduced when the high amount of WIP was added. The presence of Fe and Na_2_[Fe(OH)_4_] chemically reduced the strength. These products hindered the dissolution of the Al^4+^ and Si^3+^ ions at the FA particle’s surface, which could lead to an insufficient geopolymeric reaction. However, only in WGP samples, the higher amount of WIP contributed more compressive strength because of the increased density in the samples.

## 4. Conclusions

WIP has the potential as a filler in high-calcium FA geopolymer. This waste enhance the geopolymer properties. WIP played a significant role in heat retention after the heat curing process and provided the longer heat curing time under the heat holding situation. The addition of WIP in geopolymer increased the bulk density of the samples; however, compressive strength was lower due to the grain-shape of WIP than the control since the impurity in WIP could affect the geopolymerization reaction; however, the flexural strength significantly increased because of the interlocking of Fe clusters and its grain shape and rough surface. The occurrence of Na_2_[Fe(OH)_4_] and zeolite phases was found in the geopolymer’s matrix. The optimum content of WIP in high-calcium FA geopolymer was suggested as 20%, obtaining the 28-day compressive strength of 50.5 MPa, which was approximately 76% of the strength of the control. This sample also yielded the high flexural strength of 8.5 MPa, which was almost three times that of the conventional geopolymer.

## Figures and Tables

**Figure 1 materials-14-02515-f001:**
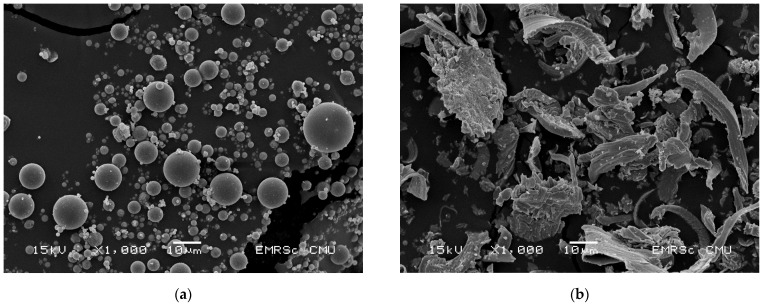
Microstructure of materials by SEM: (**a**) FA at 1000× magnification, (**b**) Waste iron powder at 1000× magnification.

**Figure 2 materials-14-02515-f002:**
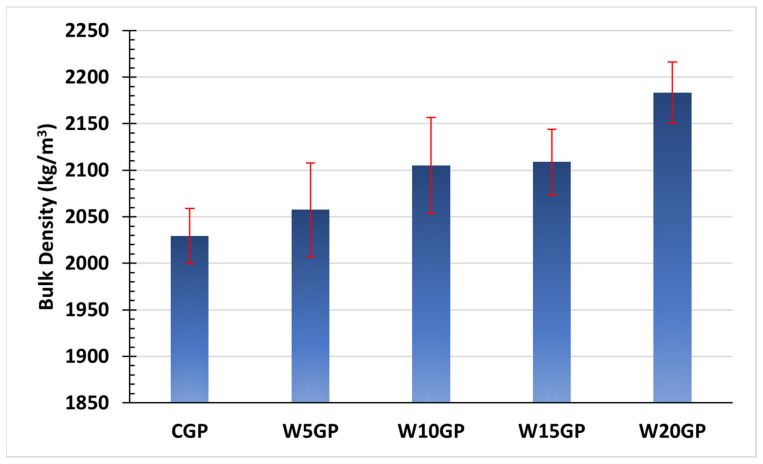
Bulk density of 8M NaOH, SS/SH of 1.00 (curing temperature of 60 °C).

**Figure 3 materials-14-02515-f003:**
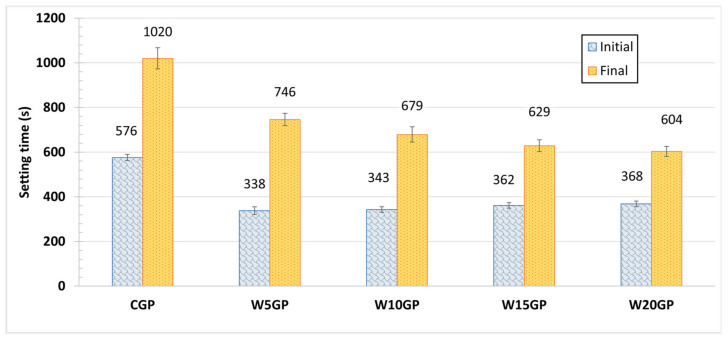
Setting times of CGP and WGP samples.

**Figure 4 materials-14-02515-f004:**
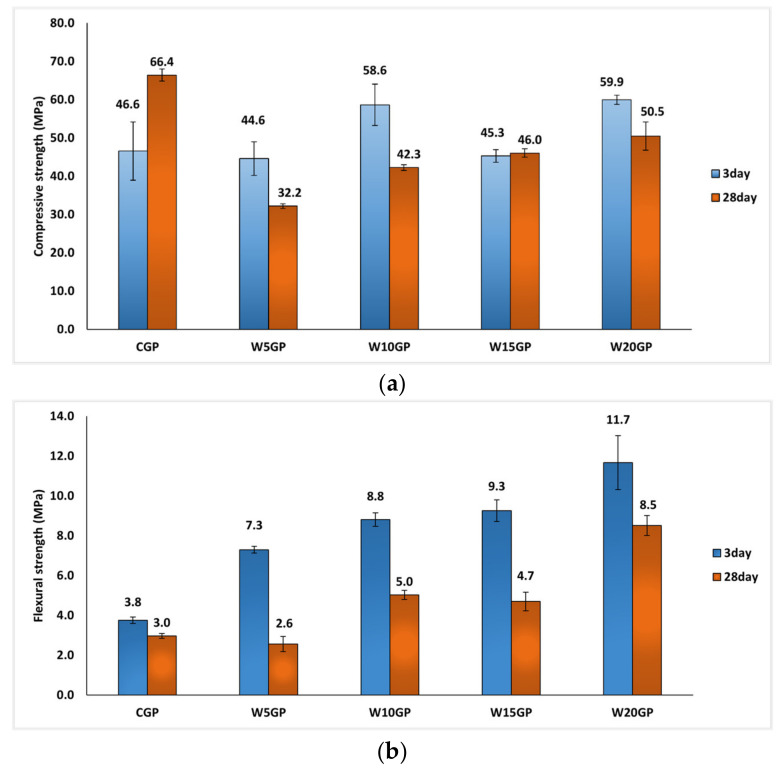
(**a**) Compressive and (**b**) flexural strengths of a sample at 8 M NaOH, SS/SH = 1.00 (heat curing temperature of 60 °C).

**Figure 5 materials-14-02515-f005:**
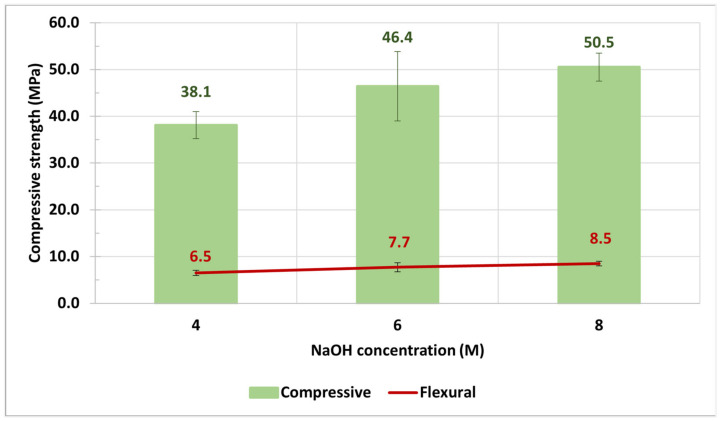
Effect of NaOH concentration on the W20GP sample at 28-day curing time (heat curing temperature of 60 °C) with SS/SH = 1.00.

**Figure 6 materials-14-02515-f006:**
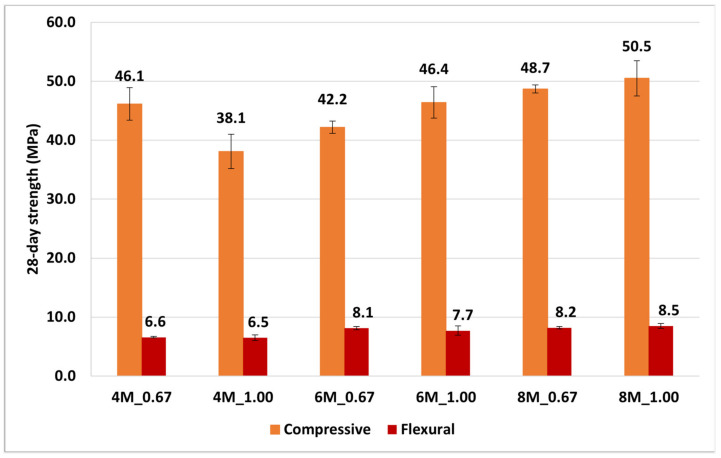
SS/SH ratio effect on 20%WIP at 28-day curing time.

**Figure 7 materials-14-02515-f007:**
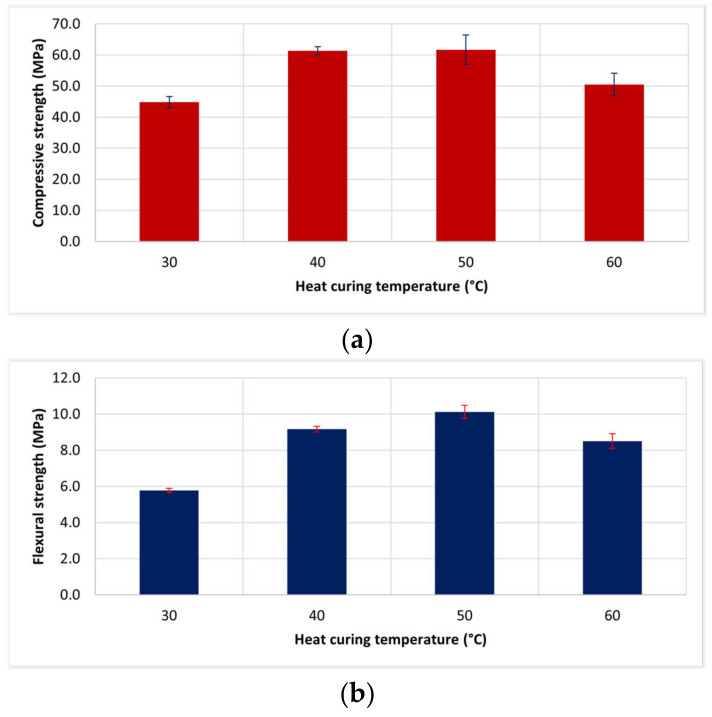
Effect of heat curing temperature on (**a**) compressive and (**b**) flexural strengths of W20GP at 28-day age with the solution of NaOH 8 M, SS/SH = 1.00, and L/B = 0.45.

**Figure 8 materials-14-02515-f008:**
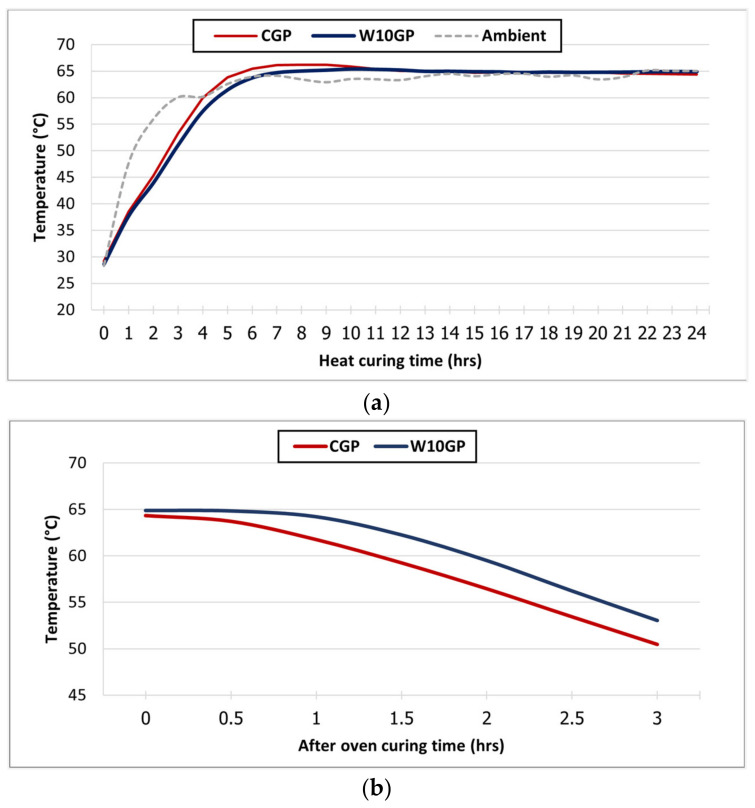
Temperature monitoring of (**a**) 24 h during the heat curing process and (**b**) 3 h after the heat curing process.

**Figure 9 materials-14-02515-f009:**
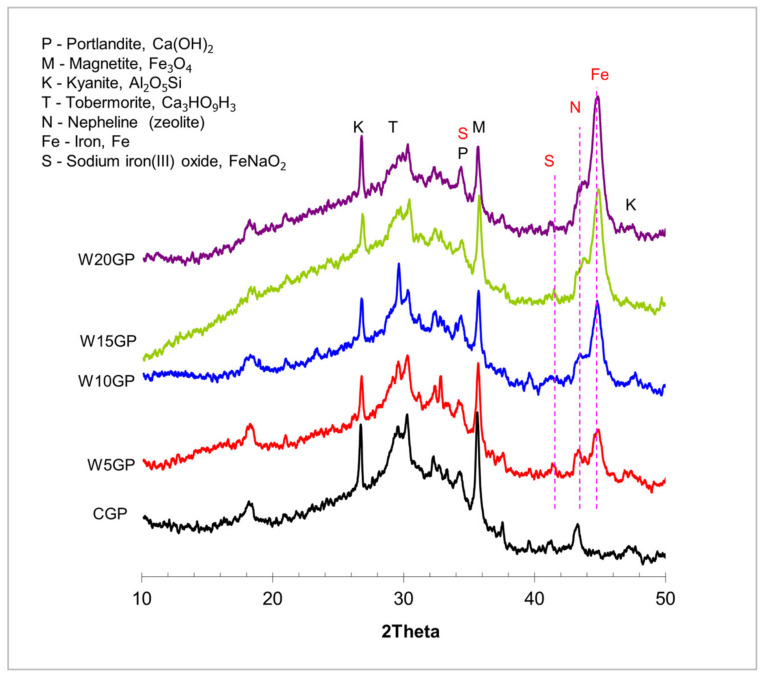
XRD patterns analysis of 28 day–aged CGP and WGP samples.

**Figure 10 materials-14-02515-f010:**
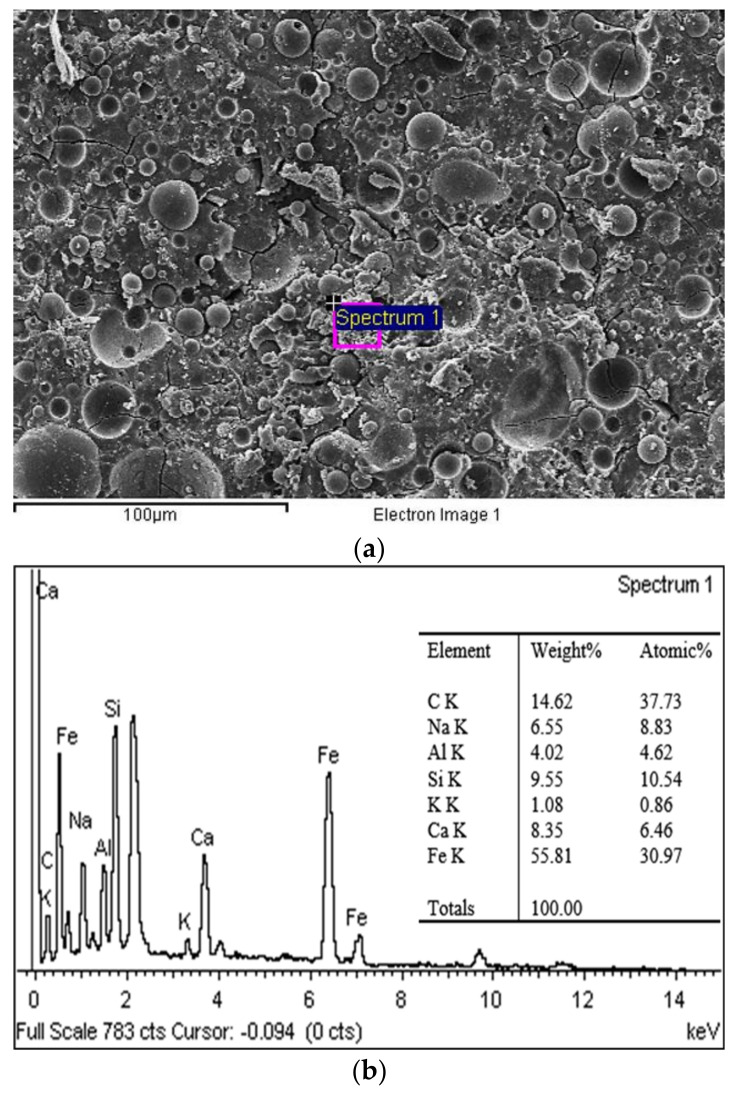
(**a**) SEM and (**b**) EDX results of W20GP at 28-day curing time.

**Figure 11 materials-14-02515-f011:**
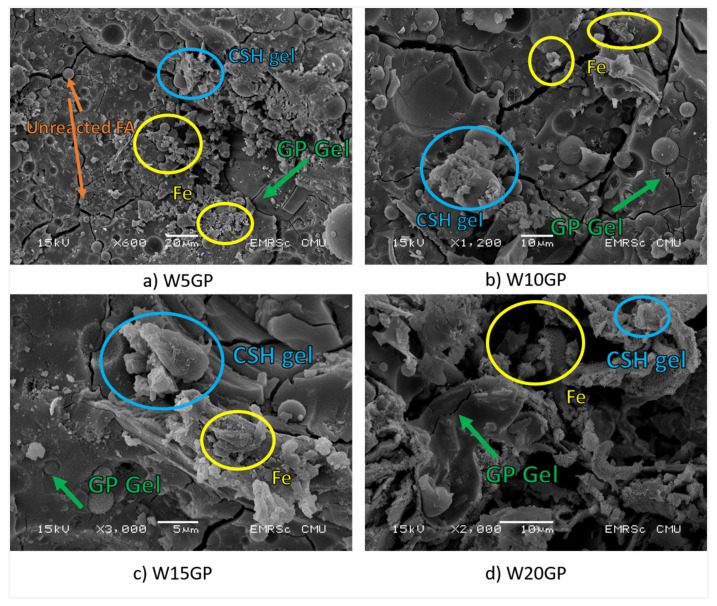
SEM figures of Na_2_[Fe(OH)_4_] existence. (**a**) W5GP at 600× magnification (**b**) W10GP at 1200× magnification (**c**) W15GP at 3000× magnification (**d**) W20GP at 2000× magnification.

**Table 1 materials-14-02515-t001:** Oxide compositions of Mae Moh fly ash and WIP.

Component	Fly Ash	WIP
Al_2_O_3_	8.52	-
SiO_2_	18.17	1.96
Fe_2_O_3_	29.85	95.95
CaO	31.41	-
SO_3_	8.43	-
K_2_O	2.47	-
TiO_2_	0.58	-
SrO	0.29	-
MnO	0.28	0.53
Cr_2_O_3_	-	1.56

**Table 2 materials-14-02515-t002:** Mixing details of all samples.

Mixture	L/B	%FA	%WIP	FA (g)	WIP (g)	NaOH (M)	SS/SH Mass Ratio	Heat Curing Temp. (°C)	Temp. Monitoring	Testing Age (days)
CGP	0.45	100	0	1500	0	4, 6, 8	0.67, 1.00	30, 40, 50, 60	✓	3, 28
W5GP	0.45	100	5	1500	75	4, 6, 8	0.67, 1.00	60	-	3, 28
W10GP	0.45	100	10	1500	150	4, 6, 8	0.67, 1.00	60	✓	3, 28
W15GP	0.45	100	15	1500	225	4, 6, 8	0.67, 1.00	60	-	3, 28
W20GP	0.45	100	20	1500	300	4, 6, 8	0.67, 1.00	30, 40, 50, 60	-	3, 28

**Table 3 materials-14-02515-t003:** Hump area under the peak between 20° and 37° 2θ.

Samples	Area (a.u.)
CGP	4110
W5GP	4032
W10GP	3644
W15GP	3346
W20GP	3005

## Data Availability

The data presented in this study are available on request from the corresponding author.

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
