# Peer review of "Characteristics of Waste Iron Powder as a Fine Filler in a High-Calcium Fly Ash Geopolymer"

_materials, 2021, doi:10.3390/ma14102515_

Round 1

Reviewer 1 Report

  1.     Does the manuscript present a specific, easily identifiable advance in knowledge?  Is it applicable and useful to the profession?

Yes this paper provides useful information on using waste iron powder to aid in giving geopolymers additional flexural strength.

  1.     Has the information already been published elsewhere, either wholly or in part? 

Not that I can tell.

  1.     Is the subject matter within the scope of the journal?  Or is it better suited to another journal?

Yes this fits well in the journal and publication type.

  1.     Do the title and abstract accurately describe the contents?  Does the abstract include all of the main findings of the study?

Now the title did not adequately describe the contents. No research was performed to describe the environmental friendly component of the title.

  1.     Is the review of literature limited to that framing the new knowledge?  Are all references pertinent and complete?

Most references are present. There seems to be a lack of references to alkali activated iron geopolymers which are directly related to this paper.

  1.     Is the methodology sufficiently well explained that someone else knowledgeable about the field could repeat the study? 

Yes this could be replicated easily.

  1.     Is each figure and table necessary to the understanding of the conclusions?  Can any be omitted without compromising the paper's message?

Figure 2 is not needed. Figures 11-13 should be re-evaluated as they probably are not all needed.

  1.     Are the results soundly interpreted and related to existing knowledge on the topic?

No there are significant misinterpretations of the data. The authors need to perform additional research in understanding ferro-sialate-based geopolymers and their related chemistry.

  1.     Are the conclusions sound and justified?  Do they follow logically from data presented?

No there are gaps that seem to arbitrarily draw conclusions from a lack of data rather than a preponderance of data they specifically collected.

  1.    Do all elements of the manuscript relate logically to the study's statement of purpose?

Most components relate to the study. However the authors indicate the hypothesis that WIP will aid in geopolymerization is based on the thermal conductivity of WIP. The problem is that the actual study only measures 2 of the 5 mixture types so no conclusion can be made related to this part of the hypothesis.

  1.    Can the paper be shortened without compromising its message? 

For the most part no. Some minor edits could be made to make this a bit more concise but overall this is well written.

Edits:

General comments:
NaOH concentration less than 10M seems small. Why was this chosen?

2.2.4 Section. Your hypothesis seems to hinge on conductivity of heat into the material but in your study, you only monitor your control and one type of specimen. Why is that?

Figure 3. Shouldn’t this be normalized in some way considering the specific gravity of WIP is about 3 times that of the fly ash? Normally we think of bulk density as a good indicator of consolidation and polymerization as good polymerization will result in a higher efficiency of unit weight but here I can’t see that information because the specific gravity influence so greatly overwhelms the results. I believe this is why the 3day strength for W10GP is so close to W20GP and better than W15GP.

When were the bulk density measurements taken? 3 days? 28 days?

Major concern here in Equation 1: if WIP is participating in the reactions, is it then still a filler? The inclusion of WIP immediately lowers the initial set time and clearly is having a chemical reaction as both initial and final are significantly lower than the control. I would argue that this entire section from line 185-195 is incorrect and the reality is not in the addition of sodium in the system but the interaction of iron hydroxide in the bridging of silane monomers to form some sort of ferro-sialate-based geopolymer. This implies that your assumption that the WIP is a filler is no longer correct and is in fact part of the polymerization process. This ought to be addressed.

Line 234. Incorrect, the grain size of the WIP is too small to adequately provide flexural strength if no chemical reaction is happening.

XRD should have been run much slower to improve the signal to noise ratio. The most important component of this figure is the amorphous hump. That is where all the geopolymer information is and should have been compared between samples.

SEM images show poorly integrated fly ash probably due to low molarity of NaOH.

The authors should include error bars on all graphs as this would likely help show statistical differences between samples. Replicates should be present. If not, this is not a publishable study.

General writing comments:

Please remove ambiguous language such as “it”, “this”, “that”, “these”, etc.

Please simplify your sentence structure to remove and minimize comma use.

Author Response

Please see the response to reviewer comments in the attached file.

Reviewer 2 Report

This paper studies the Application of waste iron powder as a fine filler in a high-calcium fly ash geopolymer: an environmentally friendly material. My comments are as follows:

Abstract

  1. A bit lengthy, please short it.

Introduction

  1. Line 43, ',[1-3]' should be '[1-3],'

Material and Methodology

  1. Line 117, please provide the oven temperature.
  2. Line 119, the ratio is by weight or volume or molecular?
  3. line 152, 'cm3' instead of 'cm3'

Results and Discussion

  1. Figures 3-8, pls provide the error bar.
  2. Figure 8 (a), x-label is a bit too dense.

Author Response

(The authors gave the same response as above.)

Reviewer 3 Report

1) Please consider improving Introduction section with more in depth analysis of what has been already achieved in the same or similar field.

2) There are no standard deviations or upper and lower limits of all results presented in the article. Please consider revising it.

3) What the red and blue lines in Figures 8a and 8b mean? The points at different temperatures are average values or individual values? If average values, then please do not show any lines. Please an effort in explaining what you wanted to show with such expression of the results.

4) Reference list does not conform the requirements of the journal.

5) Also suggesting to improve Results and Discussion section with some comparisons with other authors works, observations...

6) If you use fly ash, some leaching from the samples tests should be done because this additives is a pollutant. What happens with samples after immersion in water? How much of fly ash are immobilized and how much of them escapes from the sample?

Author Response

(The authors gave the same response as above.)

Round 2

Reviewer 2 Report

The author has addressed all the comments

Reviewer 3 Report

Authors have provided answers to all my questions.